# Molecular Assessments, Statistical Effectiveness Parameters and Genetic Structure of Captive Populations of *Tursiops truncatus* Using 15 STRs

**DOI:** 10.3390/ani12141857

**Published:** 2022-07-21

**Authors:** Rocío Gómez, Rocío M. Neri-Bazán, Araceli Posadas-Mondragon, Pablo A. Vizcaíno-Dorado, Jonathan J. Magaña, José Leopoldo Aguilar-Faisal

**Affiliations:** 1Departamento de Toxicología, Centro de Investigación y de Estudios Avanzados del IPN (CINVESTAV-IPN), Mexico City 07360, Mexico; mrgomez@cinvestav.mx; 2Laboratorio de Medicina de Conservación, Escuela Superior de Medicina-Instituto Politécnico Nacional (ESM-IPN), Mexico City 11340, Mexico; rocionerib@gmail.com (R.M.N.-B.); aposadasm@ipn.mx (A.P.-M.); 3Laboratorio de Medicina Genómica, Departamento de Genética, Instituto Nacional de Rehabilitación-Luis Guillermo Ibarra-Ibarra (INR-LGII), Mexico City 14389, Mexico; pavizcaino@inr.gob.mx; 4Departamento de Bioingenieria, Escuela de Ingeniería y Ciencias, Tecnologico de Monterrey-Campus Ciudad de México (ITESM-CCM), Mexico City 14380, Mexico

**Keywords:** *T. truncatus*, conservation programs, microsatellites, gene-flow, dolphins, population genetics

## Abstract

**Simple Summary:**

The bottlenose dolphins are one of the most used species in entertainment, assisted therapy, education, and research on welfare. However, their maintenance in captivity requires powerful and sensitive tools for preserving their diversity. The number of genetic markers for this purpose remains controversial, restraining the marine species’ genetic diversity determination. We aimed to select 15 hypervariable molecular markers whose statistical parameters were made in 210 captive dolphins from 18 Mexican centers to support their usefulness. The proposed set of markers allowed us to obtain a genetic fingerprint of each dolphin. Additionally, we identified the structure of the captive population, analyzing the groups according to the capture location. Such characterization is key for maintaining the captive species’ biodiversity rates within conservation and reintroduction programs. However, these 15 genetic markers can also be helpful for small- isolated populations, subspecies and other genera of endangered and vulnerable species.

**Abstract:**

Genetic analysis is a conventional way of identifying and monitoring captive and wildlife species. Knowledge of statistical parameters reinforcing their usefulness and effectiveness as powerful tools for preserving diversity is crucial. Although several studies have reported the diversity of cetaceans such as *Tursiops truncatus* using microsatellites, its informative degree has been poorly reported. Furthermore, the genetic structure of this cetacean has not been fully studied. In the present study, we selected 15 microsatellites with which 210 dolphins were genetically characterized using capillary electrophoresis. The genetic assertiveness of this set of hypervariable markers identified one individual in the range of 6.927e^13^ to 1.806e^16^, demonstrating its substantial capability in kinship relationships. The genetic structure of these 210 dolphins was also determined regarding the putative capture origin; a genetic stratification (*k* = 2) was found. An additional dolphin group of undetermined origin was also characterized to challenge the proficiency of our chosen markers. The set of markers proposed herein could be a helpful tool to guarantee the maintenance of the genetic diversity rates in conservation programs both in *Tursiops truncatus* and across other odontocetes, Mysticeti and several genera of endangered and vulnerable species.

## 1. Introduction

Delphinidae (dolphins and killer whales) is a diverse family including at least 12 genera and 37 species [1]. The bottlenose dolphin (*Tursiops truncatus*, Montagu, 1821) is among the most mobile species, presenting a broad distribution in the different oceans and polar coastal waters [2]. This specie exhibits size variation related to feeding, with a length ranging from 2.5 m to 3.8 m [3]. Regarding the depth of its dives, it depends on the hour, being greater than 450 m at night and tending to be shallow (50 m) during the day [4].

*T. truncatus* has been characterized depending on its distribution in coastal (>20 m isobath), continental shelf (between 20–200 m depth), inshore (i.e., bays, estuaries, and sound), and oceanic (>200 m depth) waters [5]. Each assignment shows morphological, demographic, spatial, temporal, and genetic differences [6]. Given its social capacity, *T. truncatus* is one of the most used animals in the recreational industry and assisted therapy, being one of the most widely distributed specimens in captivity [7]. In addition, research programs have included specimens living as captives to enrich the knowledge about their social lives, communication, cognitive development, natural ecology, and feeding habits, among others [8,9,10].

In agreement with the Mexican “*Ley General de Vida Silvestre*” (article 60 Bis), any species of marine mammal must not be subject to extractive use, whether for subsistence or commercial purposes [11]. Nonetheless, the Mexican aquariums and zoos still house many dolphins. Of note, this law did not prohibit scientific research and education programs, providing a unique opportunity to understand their physiology, cognitive capacities, and risk of illness [9,12,13,14]. The research and educational programs have generated scientific and robust information about the reproductive biology of these and other cetaceans with significant benefits for the conservation of wild populations [9]. Such investigations have facilitated reproductive assistance through biobank generation, which could help prevent the extinction of vulnerable species [9]. However, genetic and phenotype diversity maintenance is imperative to ensure the adaptive potential and persistence (long-term) in conservation and sustainability [15].

Despite its low-cost, morphological identification could be a subjective feature. To promote the most effective protection, the characterization of the genetic population architecture (diversity and structure) is crucial for efficient and effective preservation practices [6]. Thus, the first step in the reintroduction programs is to determine the genetic structure, especially in those species lacking physical dispersal barriers and coming from different geographic origins [16,17]. Thus, genetic tools are cornerstones that highlight the reproductive potential and delineate the diversity and phylogenetic relationships [18].

Microsatellites or short tandem repeats (STRs) are multiallelic markers appropriate for population genetic studies and pedigree analyses. These hypervariable molecular markers may be amplified with modest DNA concentrations, being a cheaper process than next-generation sequencing methods [19]. However, the informativeness degree and the set number of decision-making markers supporting the conservation programs and protecting biodiversity have been poorly studied. Mainly, the parentage identity allows for identifying genetic differentiation levels, a fundamental aspect of reproduction strategies’ success [20,21]. Although many articles support the use of STRs, only a limited number have reported the probability of identity [22,23].

In this study, the genetic diversity and population structure of 210 captive dolphins (*T. truncatus*) from 18 Mexican dolphinariums were characterized using 15 STRs. This set of markers allowed us to genetically distinguish each dolphin with remarkable efficiency (discrimination power ≥ 0.999), evidencing its robustness at a low cost. Our findings validated using these 15 STRs to evaluate the probability of identity and characterize the genetic structure and diversity of genome banks used in artificial insemination, which has been poorly described.

## 2. Materials and Methods

### 2.1. Study Population

The present study was conducted in agreement with the Federal Attorney for Environmental Protection (PROFEPA, initials in Spanish) in Mexico City to carry out joint actions for the diagnosis, surveillance, prevention, and control of diseases in any animal species whose survival is threatened. The study was approved by the Research Committee of the *Escuela Superior de Medicina* from the *Instituto Politécnico Nacional* (ESM-IPN, initials in Spanish).

Field research was carried out in 18 Mexican centers of dolphin captivity, distributed in eight states with tourist impact (Appendix A). Two hundred and ten blood samples were obtained from the caudal fins of *T. truncatus* living captive. The capture locality was collected in 177 individuals from expedients, grouping these specimens according to this information. Nevertheless, its demographic history was unknown. Out of 210 dolphins, 104 were putatively assigned to Mexican waters (MD; 92 of these were obtained from the California Gulf, MD1, and 12 from the Gulf of Mexico, MD2). Likewise, 46 dolphins were assumed to be from Cuban territorial waters (CD), 14 from southern Australian waters (AD), and 5 from Japanese (JD) waters. The rest of the dolphins (41) had an unknown geographic origin (undetermined dolphins; UD) (Appendix A).

### 2.2. Genotyping Analysis

Genomic DNA was isolated using the MagNA Pure LC Total Nucleic Acid Isolation kit (Qiagen, Hilden, NRW, Germany) in the MagNA Pure LC automated equipment (Roche, Germany). DNA purity (λ_260_/λ_280_) and concentration (ng/μL) were evaluated with Nano-Drop 1000 (Thermo Fisher Scientific, Suwanee, GA, USA). DNA integrity was checked with agarose (Sigma Aldrich, St. Louis, MO, USA) gel 0.8% stained with ethidium bromide (Sigma Aldrich, St. Louis, MO, USA).

Samples were genotyped with 15 microsatellite *loci* previously reported (Table 1). These 15 STRs were chosen based on available information from earlier population genetic studies, and an observed heterozygosity of at least 0.600 was considered [24,25,26,27,28,29]. PCR was performed in a total volume of 10 μL containing 200 nM of each primer, 35 ng DNA template, 200 mM of dNTPs, 1 × reaction buffer, 1.5 mM of MgCl_2_, and 0.03 U of *Taq* DNA polymerase (Roche Diagnostics GmbH, Mannheim, BW, Germany). The amplification conditions consisted of 35 cycles of an initial denaturation at 94 °C for 3 min, followed by a denaturation at 94 °C, annealing (depending on each primer; Table 1) and extension at 72 °C for 1 min. Thermal cycling was carried out on an Applied Biosystems Thermal Cycler (Veriti 96 Well Thermal Cycler, Carlsbad, CA, USA). Primers at the 5′- end of forward were labelled with the 6-FAM fluorescent dye. Allelic discrimination was carried out in a 3730xl DNA analyzer (Applied Biosystems, Carlsbad, CA, USA). Capillary electrophoresis conditions were 15 kV at a constant temperature of 60 °C with POP-7 polymer for 24 min (Applied Biosystems, Carlsbad, CA, USA); GeneScan^TM^ 500 TAMRA^TM^ dye was used as the internal size standard (Applied Biosystems, Carlsbad, CA, USA). Allelic assignment was performed with the GeneScan v3.2 software (Applied Biosystems, Carlsbad, CA, USA).

### 2.3. Statistical Analysis

The obtained genotypes were subjected to statistical analyses, including allele and genotype frequencies, number of alleles (*k*), and expected (He) and observed (Ho) heterozygosity using Arlequin v3.5.2.2; (Berne, Switzerland) [30]. The number of effective alleles (Ne) and Shannon’s index (I) were determined with GenAlEx v6.5; (Canberra, Australia) [31]. Hardy–Weinberg’s (HW) expectation by Weir and Cockerham’s *F* statistics (*F_IS_*) was estimated with Gènetix v4.05.2; Gènetix team (Montpellier, France) using 10,000 permutations. Linkage disequilibrium (LD) was estimated with FSTAT v2.9.3.2; (Lausanne, Switzerland) using 105,000 permutations [32].

Null alleles and large dropouts were checked for each *locus* with Genepop v4.6 and FreeNA and confirmed in Micro-Checker v2.2.3; (Hull, UK) [33,34].

Matching probability (MP), discrimination power (PD), and the polymorphic information content (PIC) were assessed using PowerStats v1.2; Promega CORP.(Wisconsin, USA) [35].

The genetic structure was inferred with Structure software v2.3.3; (Oxford, UK) using 100,000 burn-in and 700,000 after burn-in interactions [36]. Five independent runs were performed for each K-value (1 to 10). The output file was analyzed to find the most probable K-value with the Structure Harvester program [37]. The genetic distances (*F_ST_* values) were assessed with Arlequin v3.5.2.2; (Berne, Switzerland) (10,000 permutations) and adjusted by a false discovery rate test in R-software; (Tel Aviv, Israel) [30,38]. These values were visualized in a multidimensional scale plot (MDS) with SPSS v11; IBM CORP. (New York, NY, USA) [39].

## 3. Results

### 3.1. Microsatellite Diversity Parameters and Paternity Effectiveness

Overall, all *loci* presented at least seven different alleles with a mean PIC of 0.766. The most informative markers were Ttr63 (*k* = 26), EV37 (*k* = 24), and MK06 (*k* = 18), whereas TEXVET07 (*k* = 7), MK09 (*k* = 8), Ttr58 (*k* = 8) and KWM02 (*k* = 9), were the least informative (Appendix A). Of note, the allele frequency distributions exhibited bi- and trimodal distributions.

MP and PD are summarized in the Appendix A. Even though some *loci* individually showed low diversity, globally, all 15 STRs were highly informative, identifying one individual in billions of dolphins (1.141 × 10^−16^) with a highly efficient PD (PD ≥ 0.999).

### 3.2. Null Alleles, Hardy–Weinberg Expectation, and Linkage Disequilibrium

Given the high inbreeding values (*F_IS_* mean = 0.198; range: 0.058–0.423), we determined and confirmed null alleles using two different programs. The null allele frequencies ranged from 0.023 to 0.185 with wide confidence intervals (Appendix A), except in Ttr11, Ttr58, and TEXVET5. None of these *loci* presented evidence of large allele dropout.

Regarding the HW expectations, we found a remarkable departure (HWD) in 13 out of 15 *loci* even after Bonferroni’s correction (*p* ≤ 0.0034). The HWD was related to a homozygosis excess (*F_IS_* > 0). A significant number of *loci* pairs exhibited LD adjusting the *p*-value for 5% (*p* ≤ 0.00047) and 1% (*p* ≤ 0.000095) of the nominal level (Appendix A).

### 3.3. Population Structure

*F**ST* values were also estimated to compare the variation among the populations; UD was included in this analysis (Table 2). High population differentiation was presented between all groups, showing them well-separated from each other. The most separated dolphin groups were CD vs. JD, followed by CD vs. AD. Lowest but significant differences were found between CD and MD; MD and UD showed a nuance genetic distance and in turn, a nonsignificant genetic difference.

Given the bi and trimodal distributions described before, we performed a PCoA analysis using the putative origin as a criterion. We created a genetic structure using all individuals to delve into this possibility; the Bayesian method revealed two inferred genetic clusters (*K* = 2; *p* = 1; Figure 1).

These data were used to check whether the putative geographic origin could suggest certain similarities with those inferred with structure. The MDS plot depicted similarities among MD, CD, and UD putative origin populations. MD2 and CD exhibited a marginal similarity (*p* = 0.031). Nevertheless, MD1 and MD2 did not differ significantly regarding UD (Figure 2). Likewise, JD presented a marginal difference with MD1. These data suggest that the putative MD1, MD2, CD, JD, and UD could form the first subpopulation. By contrast, AD was set apart from the rest of the groups.

## 4. Discussion

Genetic tools are indispensable for maintaining diversity with implications for reintroduction and conservation of endangered and vulnerable species. Statistical parameters and population genetic studies are crucial to validate the robustness of the markers selected; a critical point for their choice [40]. Particularly, the effectiveness and the informative degree of microsatellites used in the cetacean characterization have not been fully studied. Our particular interest was to know the effectiveness of genetic parameters and kinship relationships to guarantee biodiversity and implement identification strategies. The present study reports the genetic structure, the DNA fingerprint effectiveness, and the genetic diversity patterns of 210 *T. truncatus* individuals living captive from 18 Mexican centers using 15 STRs.

Previous studies have used multiple genetic markers for identification purposes [24,25,26,27,28,29]. Nonetheless, the sample sizes have been modest, whereas the statistical parameters have been restricted to the number of alleles, heterozygosity, and PIC. Such parameters are not informative and lack the effectiveness guideline, increasing the research costs. Knowledge of the statistical effectiveness parameters is crucial for several disciplines. Our study validated the statistical parameters of a set of 15 markers, reporting its specifications regarding kinship relationships and discrimination power. The set of markers proposed herein exhibited a remarkable discrimination capability (1.141 × 10^−16^), efficiently in distinguishing one individual in a range of millions of dolphins. Such effectiveness was comparable with the data obtained from 19 STRs, although more cost-effective [22]. These features were the greatest strengths of our study, reinforcing its use in several disciplines. However, the prior studies on these microsatellites were essential for our study [24,25,26,27,28,29].

Of note, the probability of identity was obtained assuming random mating. Notably, the *F_IS_* values found suggest “inbreeding”, impacting the genetic variance and covariances between relatives [41]. Hence, this proficiency could be nuanced, depending on genetic closeness, and should be interpreted with caution [42]. Such “inbreeding” might reflect certain adaptations to captivity [43,44]. The human-induced selection over short time frames could also be causal to the loss of alleles and genotypes considered beneficial in wildlife [45]. Furthermore, using these 15 STRs could be extremely valuable in conservation and management strategies of endangered and vulnerable species, which requires several *loci* and molecular assessments to monitor the diversity [24]. Given that these STRs are highly conserved across odontocetes, their employment could encompass several genera, thereby adding to the scope of our findings. Several markers of the present study have been amplified in protecting species belonging to genera such as *Balaena*, *Balaenoptera*, *Delphinapterus*, *Eschrichtius*, *Globicephala*, *Grampus*, *Megaptera*, *Stenella*, *Orcinus* and *Pontoporia*, to mention a few [24,27,28,46].

Even though *T. truncates* is not considered an endangered species, these markers allowed us to determine its genetic structure by identifying two well-defined populations [8]. Such stratification could be associated with the capture habitat and possibly with the different ecotypes (i.e., inshore and outshore) [20,47,48,49,50]. The HWD and the remarkable LD found in the whole population are signatures of population stratification, confirmed through several analyses. Especially, the inshore ecotype has been reported in the Atlantic and Caribbean dolphins [47,49,51,52]. Coastal and offshore ecotypes have been documented in the Gulf of Mexico, being the inshore ecotype almost six times more prominent than the other one [49,53]. Genetic and isotopic differentiation have also been reported between the Gulf of California and the west coast of Baja California [54]. Thus, it is likely that some ecotype was shaping the genetic architecture of the samples, with the offshore one being the most probable given its high frequency [55,56]. Our data are discrepant from prior studies in other regions where low genetic diversity has been reported [57]. Such differences reinforce the necessity of genetic studies and a validated set of markers to expand the knowledge about diversity within *Tursiops*. These studies should also consider the environmental changes that have altered the genetic structure, suggesting that geography is only one of the different aspects contributing to genetic differentiation [16].

On the other hand, the knowledge of the genetic structure has been associated with the reintroduction facility, being successful where both populations (reintroduced and acceptor) share genetic similarities [17]. Diversity generation is also considered to minimize the inbreeding rates and ensure persistence [44,58]. In turn, the genetic structure is a cornerstone to determining survival and thriving [17]. Thus, before the reintroduction, random sampling from some wild individuals belonging to the putative acceptor populations should be performed to identify the genetic profiles [58]. In this setting, the proposed set of markers could be particularly valuable. Despite their utility, these 15 STRs should be accompanied by markers such as mitochondrial DNA (mtDNA) and other, more cost-effective markers such as the visual cues provoked by the damage to the tip of the dorsal fin [23,51]. Particularly, mtDNA is useful for identifying the migratory patterns and even identifying species and subspecies [22,59]. Although the markers employed herein exhibited a high identification ability, genome-wide genetic variation has been considered the best approach to maintaining the species’ biodiversity [60]. Nonetheless, this technology’s high costs are still prohibitive to developing countries such as Mexico. Alternatively, this set of markers, along with radio frequency identification and the permanent scars, might help avoid impersonation and illegal trafficking of these and other marine mammals.

Regarding the remarkable diversity found, we did not rule out that it could be related to other species such as *Tursiops aduncus* (*T. aducus*) that could have been confused with **Tursiops truncatus* truncatus* (*Tursiops t. truncatus*). The genetic characterization using STRs and mtDNA has shown the presence of *T. aducus* in coastal waters of southern Australia [61]. Unfortunately, the absence of non-nuclear genetic markers (i.e., control, D-loop, and Cytochrome regions) to clarify this doubt limited our study [22,49,62]. Another approach could be the analysis of genetic distances and their visualization in an MDS plot in Australian *Tursiops t. truncatus* and *T. aduncus*. Nonetheless, not all authors share their raw data or analyze the same *loci*, hindering such analysis. In this setting, the “inbreeding” exhibited by the studied populations was inconsistent with the highest diversity values and the greatest number of alleles, reinforcing this possibility.

Although genetic characterization is a key topic in the conservation and reintroduction process, it is only one of the diverse variables that should be considered [17,43,63]. Variables such as habitat, climate requirements, and management of pathogens carried by the reintroduced species, among others, must be considered in the reintroduction planning and individual management [63]. As regards the captive *T. truncatus*, their reintroduction is not feasible, mainly because they are not vulnerable populations and are dependent on humans. Foraging strategies and social relationships have been modified in captive dolphins due to human dependence [13]. In such circumstances, these cetaceans look forward to approaching vessels and people with subsequent implications such as injury from ingestion or even death [13]. Thus, reintroduction biology integrates multiple fields of study [58].

As mentioned before, the new regulations (i.e., article 60 Bis) did not prohibit scientific research and education programs [11] Thus, the collected information from ex situ dolphins could be extrapolated to wildlife specimens, promoting their welfare [8,9]. Research programs in high-income countries have provided substantial information about behavioral, cognitive, and physiological attributes, disease risk, and even the education of visitors, impacting their sensitivity and appreciation for these mammals [9,14,64,65]. Specifically, our markers could contribute positively to the characterization of biobanks and, subsequently, to reproductive assistance programs [9]. Hence, research programs represent a relevant opportunity to expand our knowledge about these and possibly other odontocetes [9,66].

## 5. Conclusions

In summary, the results presented herein validated using these 15 STRs to evaluate the probability of identity and to characterize the genetic structure with remarkable efficiency. These hypervariable markers are powerful and sensitive, enabling the genetic characterization of this and other species at a low cost, preserving the diversity amongst individuals. Nevertheless, more studies involving uniparental markers, sex determination, cranial and other morphological measurements should be determined to complete such a panorama.

## Figures and Tables

**Figure 1 animals-12-01857-f001:**
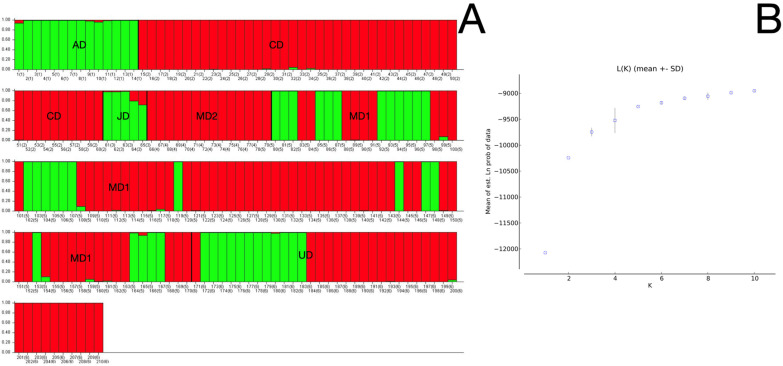
Genetic structure of the *T. truncatus* populations analyzed in this study (four putative geographic populations and one population with unknown geography). (**A**) Bar plot of the several individuals analyzed; each vertical line (*x*-axis) is a single individual with colors representing each cluster’s membership proportion. Colors (red and green) represent the subpopulations conforming to the putative geographic origin (**B**) Mean L(K) ± SD over five runs for each K value. Note: AD: Australian dolphins; CD: Cuban dolphins; JD: Japanese dolphins; MD: Mexican dolphins; UD: Unknown dolphins.

**Figure 2 animals-12-01857-f002:**
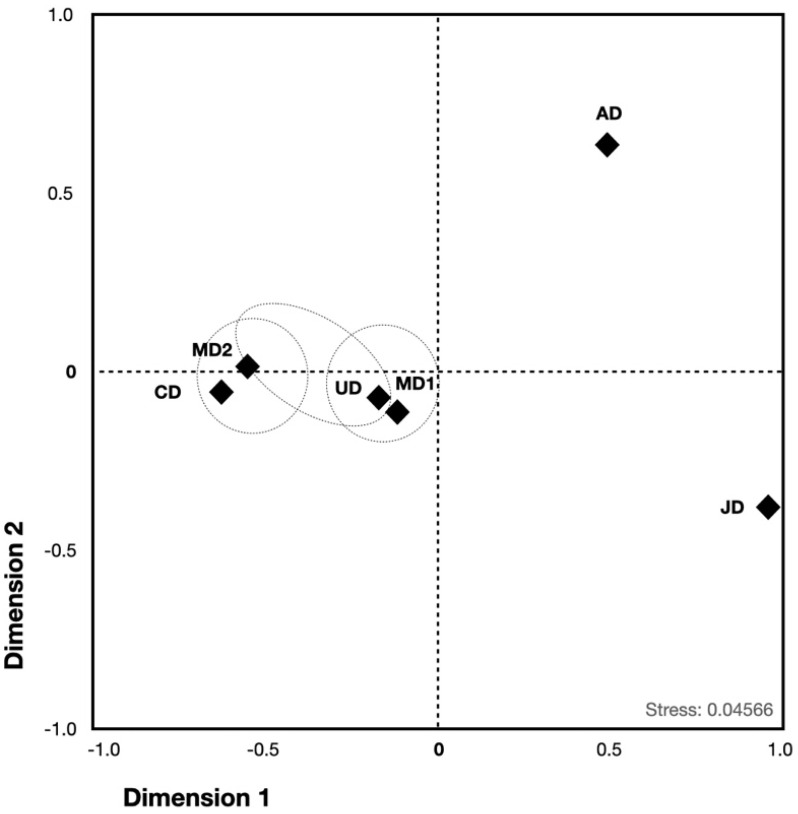
MDS plot of *R_ST_* values estimated from 15 STRs, including four *T. truncatus* geographic populations and one population with unknown geography. Diamonds represent the position of each putative population in agreement with the genetic distance whereas the dotted circle encloses those populations that share no significant genetic distances. Note: AD: Australian dolphins; CD: Cuban dolphins; JD: Japanese dolphins; MD: Mexican dolphins; UD: Unknown dolphins.

**Table 1 animals-12-01857-t001:** Sequence of 15 *loci* explored in the four putative groups of *T. truncatus*.

*Locus*	Gene Bank Access Number	Tandem Repeat	Sequence primer (5′–3′)		Allele Size	Annealing T	Reference
D08	NA	(TG)n	F	GATCCATCATATTGTCAAGTT	94–122	58	[27]
			R	TCCTGGGTGATGAGTCTTC			
EV37	NA	(AC)n	F	AGCTTGATTTGGAAGTCATGA	189–241	56	[28]
			R	TAGTAGAGCCGTGATAAAGTGC			
KWM2	NA	(AC)n	F	GCTGTGAAAATTAAATGT	138–160	47	[29]
			R	CACTGTGGACAAATGTAA			
KWM9	NA	(AC)n	F	TGTCACCAGGCAGGACCC	170–188	59	[29]
			R	GGGAGGGGCATGTTTCTG			
KWM12	NA	(AC)n	F	CCATACAATCCAGCAGTC	160–186	50	[29]
			R	CACTGCAGAATGATGACC			
MK6	AF237891	(GT)n	F	GTCCTCTTTCCAGGTGTAGCC	147–187	51	[24]
			R	GCCCACTAAGTATGTTGCAGC			
MK8	AF237892	(CA)n	F	TCCTGGAGCATCTTATAGTGGC	80–114	58	[24]
			R	CTCTTTGACATGCCCTCACC			
MK9	AF237893	(CA)n	F	CATAACAAAGTGGGATGACTCC	161–175	54	[24]
			R	TTATCCTGTTGGCTGCAGTG			
Ttr04	DQ018982	(CA)n	F	CTGACCAGGCACTTTCCAC	103–127	65	[25]
			R	GTTTGTTTCCCAGGATTTTAGTGC			
Ttr11	DQ018981	(CA)n	F	CTTTCAACCTGGCCTTTCTG	193–219	61	[25]
			R	GTTTGGCCACTACAAGGGAGTGAA			
Ttr19	DQ018980	(CA)n	F	TGGGTGGACCTCATCAAATC	182–200	61	[25]
			R	GTTTAAGGGCTGTAAGAGG			
Ttr58	DQ018985	(CA)n	F	TGGGTCTTGAGGGGTCTG	166–194	62	[25]
			R	GTTTGCTGAGGCTCCTTGTTGG			
Ttr63	DQ018986	(CA)n	F	CAGCTTACAGCCAAATGAGAG	83–149	59	[25]
			R	GTTTCTCCATGGCTGAGTCATCA			
TexVet5	AF004905	(CA)n	F	GATTGTGCAAATGGAGACA	196–216	55	[26]
			R	TTGAGATGACTCCTGTGGG			
TexVet7	AF004907	(CA)n	F	TGCACTGTAGGGTGTTCAGCAG	155–169	64	[26]
			R	CTTAATTGGGGGCGATTTCAC			

Note: NA = Not available in the Gene Bank database.

**Table 2 animals-12-01857-t002:** *F**ST* values estimated from 15 *loci* studied in the four groups of *T. truncatus*.

	AD	CD	JD	MD	UD
AD	-	**≤0.0001**	**≤0.0001**	**≤0.0001**	**≤0.0001**
CD	0.27886	-	**≤0.0001**	**≤0.0001**	**≤0.0001**
JD	0.15746	0.23717	-	**≤0.0001**	**≤0.0001**
MD	0.17679	0.05293	0.11988	-	0.25225 ± 0.0264
UD	0.20180	0.03210	0.13256	0.00965	-

Note: AD: Australian dolphins; CD: Cuban dolphins; JD: Japanese dolphins; MD: Mexican dolphins; UD: Unknown dolphins. Bold numbers represent significant *p*-values; Bonferroni’s correction *p*-value = 0.0033. Values above the hyphen represent *p*-values; those values below the hyphen represent the genetic distances.

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
