# Peer review of "Molecular Assessments, Statistical Effectiveness Parameters and Genetic Structure of Captive Populations of Tursiops truncatus Using 15 STRs"

_animals, 2022, doi:10.3390/ani12141857_

Round 1

Reviewer 1 Report

Gomez et al. present an analysis of 15 microsatellite markers as evidence for managing captive Tursiops truncatus (bottlenose dolphins) by examining >200 individuals. The paper contributes new useful information and analyses on the genetics of these animals. The authors might consider the following in improving their manuscript:

0) English grammar, word-usage, and other minor typos should be fixed throughout the text.

1) line 80. I did not understand the statement, "This 80 set of markers allowed to distinguish one individual in a range of millions to billions of dolphins with remarkable efficiency (discrimination power ≥ 0.9999999999)." First, there are not this many bottlenose dolphins on Earth, but also, this does not seem to take into account close relatives in inbred subpopulations or other phenomenon perhaps? At any rate, I think the average reader will be maybe a little confused by this statement early in the introduction. Does this estimated assume random mating within the entire species worldwide or some other odd simplifying assumptions?

2) line 95. "...wild T. truncatus living captive" is a confusing phrase? The animals are either 'wild' or 'captive' but can't really be both?

3) Figure 1. In the caption of the figure, 11 recreational centers are noted, but I did not immediately understand what the numbers in the figure correspond to. There are more than eleven dots on the map, and numbers range from 1 to 36 and there are more than eleven of these as well. Additional clarification is needed in the figure legend?

4) line 99. Given the most recent taxonomy of species for the genus Tursiops, is it 100% sure that all specimens are T. truncatus and none from T. aduncus or other putative species in the genus? Are there mtDNA data from these specimens that might confirm species IDs of the samples, in particular the Australian ones? Or, is none of this a concern? Perhaps just sequencing barcodes (mt COI) for one or two Australian specimens in the sample would confirm or refute a problem like this?

5) line 170. Does the high degree of linkage disequilibrium impact conclusions based on this dataset which may, or may not, assume linkage problems and inbreeding which was noted in this section?

6) line 218 and following paragraph. It should maybe be clarified how breeding in captivity and reintroductions from these captive individuals should proceed. Do the authors think distant individuals in terms of genetics (e.g., Mexican versus Australian dolphins) should be interbred even though they are very geographically distinct and may never mate in the wild? Or, should genetically diverse individuals like this be bred in captivity so as to lower inbreeding in captivity, etc.? Perhaps more of these issues could be expanded on a bit in the discussion section.

Reviewer 2 Report

This is a very well-written and interesting paper, which can be accepted pending minor revisions addressing the following comments:

-A reference is needed for the first sentence of the Introduction.

-Insert authorship for Tursiops truncatusin the second sentence of the Introduction and, if required by the journal, insert the reference in the list of references.

-Add maximum size and maximum recorded depth for Tursiops truncatusin the Introduction.

-Line 82: reduce the number of decimal places.

-Materials and Methods: be consistent with inserting a blank space between numbers and units of measure.

-Line 150: the period should be moved to the end of the sentence.

-Line 161: reduce the number of decimal places.

-Line 163: do not insert blank spaces before and after the n-dash.

Reviewer 3 Report

The principal flaw of the manuscript is that the results do not support its conclusions. From the title on, there is an intention to highlight the potential use for conservation and reinsertion (AKA reintroduction) of a set of selected microsatellites, but the authors fail in showing why this is so essential. There is no information (or even discussion) on the genetic structure of the wild populations that would be targeted for reintroduction, and how this new technique can make this even possible (or useful). The bottlenose dolphin is a species listed as “Least Concern” by the IUCN’s red list of endangered species, consequently, it does not have conservation needs. especially not reintroduction with is a last resource conservation tool. Trying to propose reintroduction in a non-endangered species does not have any sense from the conservation point of view. On the other hand, the methodology proposed in the manuscript is an improvement strongly based on previous genetic studies made on Tursiops truncatus (which is surprisingly not clear enough in the methods section, with a total lack of references to previous authors), and that improvement would be probably the only interesting result to be published. Strangely enough, the authors try to justify the need for an unnecessary reintroduction to highlight the importance of an improvement in a genetic method, instead of comparing the efficacy of their method with the ones previously published.

Line 18  Just referring to the use of dolphins under human care for entertainment and assisted therapy does not seem a complete description. See for example:

Miller, L. J., Zeigler-Hill, V., Mellen, J., Koeppel, J., Greer, T., & Kuczaj, S. (2013). Dolphin Shows and Interaction Programs: Benefits for Conservation Education? Zoo Biology, 32(1), 45–53. https://doi.org/10.1002/zoo.21016

Harley, H. E., Fellner, W., & Stamper, M. A. (2010). Cognitive Research with Dolphins ( Tursiops truncatus ) at Disney ’ s The Seas : A Program for Enrichment , Science , Education , and Conservation. International Journal of Comparative Psychology, 23, 331–343.

Cunningham-Smith, P., Colbert, D. E., Wells, R. S., & Speakman, T. (2006). Evaluation of Human Interactions with a Provisioned Wild Bottlenose Dolphin (Tursiops truncatus) near Sarasota Bay, Florida, and Efforts to Curtail the Interactions. Aquatic Mammals, 32(3), 346–356. https://doi.org/10.1578/AM.32.3.2006.346

O’Brien, J. K., & Robeck, T. R. (2010). The Value of Ex Situ Cetacean Populations in Understanding Reproductive Physiology and Developing Assisted Reproductive Technology for Ex Situ and In Situ Species Management and Conservation Efforts. International Journal of Comparative Psychology, 223(2009), 227–248.

Fahlman, A., Brodsky, M., Miedler, S., Dennison, S., Ivančić, M., Levine, G., … Borque-Espinosa, A. (2019). Ventilation and gas exchange before and after voluntary static surface breath-holds in clinically healthy bottlenose dolphins, Tursiops truncatus. The Journal of Experimental Biology, 222(5), jeb192211. https://doi.org/10.1242/jeb.192211

Clegg, I. L. K., & Delfour, F. (2018). Cognitive judgement bias is associated with frequency of anticipatory behavior in bottlenose dolphins. Zoo Biology, 37(2), 67–73. https://doi.org/10.1002/zoo.21400

Lima, A., Sébilleau, M., Boye, M., Durand, C., Hausberger, M., & Lemasson, A. (2018). Captive Bottlenose Dolphins Do Discriminate Human-Made Sounds Both Underwater and in the Air. Frontiers in Psychology, 9(January), 55. https://doi.org/10.3389/fpsyg.2018.00055

Giménez, J., Ramírez, F., Almunia, J., G. Forero, M., & de Stephanis, R. (2016). From the pool to the sea: Applicable isotope turnover rates and diet to skin discrimination factors for bottlenose dolphins (Tursiops truncatus). Journal of Experimental Marine Biology and Ecology, 475. https://doi.org/10.1016/j.jembe.2015.11.001

Lines 19-22 Genetic tools are critical to keeping genetic diversity in a captive population but they are not useful for control and identification of the specimens, dolphins are clearly identified by visual cues which is much more cost-effective for specimen identification than genetic markers. 

Line 30 Use reintroduction instead of reinsertion

Line 45 Use reintroduction instead of reinsertion

Line 51 The Society for Marine Mammalogy’s taxonomic committee recognizes 37 species

Lines 59-60 The provided reference does not support the prohibition on the use of marine mammals stated in the text. In fact, it does not even refer to aquatic or marine mammals, cetaceans, or even dolphins.

Lines 60-62 The authors suggest that the whole population of captive dolphins in Mexico should be “reinserted” (reintroduced) to the wild, but this is not supported by any reference, please support this with the necessary reference. 

Lines 63-84 Despite the authors referring to the need for this study for conservation and reinsertion (AKA reintroduction) it remains unclear how this new set of markers is going to be useful in the conservation of a species considered of Least Concern (Non endangered) by the International Union for the Conservation of Nature. The IUCN’s reintroduction specialist group does not support reintroduction of non-endangered species, as it would not be any conservation benefit and it could be detrimental to the species. This must be clarified and supported with scientific references.

Figure 1. ¿Does the distribution of the dolphins by state add any significant information the publication?

Lines 114-115 The authors mention that 11 of the 15 strings used for the analysis were described in the previously published analysis, this should be explained in more detail and the previous studies clearly cited, not just included in the supplementary material.

Table: For clarity table, S2 should be included in the methods section of the manuscript and not in the supplementary materials.

Lines 200 Reference [7] is a study on subjective personality in dolphins, it does not have anything to do with the statement

Lines 201-214 The discussion does not add any substantial information on how this methodology is going to help the conservation of the wild populations of dolphins in Mexico, nor how it can be applied to gain relevant information on the genetic structure of the wild populations of dolphins.

Lines 221 How can it be used in reintroduction programs of a non-endangered species according to the IUCN?

Lines 226-228 To make such a general assumption a detailed discussion on the management of the captive population in Mexico should be done. This inbreeding situation is not the case in other dolphin captive populations (EEUU or Europe for example), hence it could be the result of a particular population management problem in the country.

Lines 258-260 The conclusion is not supported by the data presented.

Round 2

Reviewer 3 Report

Response to Reviewer 3 Comments

a) The principal flaw of the manuscript is that the results do not support its conclusions. From the title on, there is an intention to highlight the potential use for conservation and reinsertion (AKA reintroduction) of a set of selected microsatellites, but the authors fail in showing why this is so essential. There is no information (or even discussion) on the genetic structure of the wild populations that would be targeted for reintroduction, and how this new technique can make this even possible (or useful). The bottlenose dolphin is a species listed as “Least Concern” by the IUCN’s red list of endangered species, consequently, it does not have conservation needs. especially not reintroduction with is a last resource conservation tool. Trying to propose reintroduction in a non-endangered species does not have any sense from the conservation point of view. On the other hand, the methodology proposed in the manuscript is an improvement strongly based on previous genetic studies made on Tursiops truncatus (which is surprisingly not clear enough in the methods section, with a total lack of references to previous authors), and that improvement would be probably the only interesting result to be published. Strangely enough, the authors try to justify the need for an unnecessary reintroduction to highlight the importance of an improvement in a genetic method, instead of comparing the efficacy of their method with the ones previously published.

  • Following this meaningful observation, we have included a conscientious discussion of the potential use of these genetic markers for the management and possible reintroduction utilities with advantages and limitations of the study (see Discussion). According to the reviewer, we discuss the low feasibility of reintroducing a low threatened species. We now referred in table 1 the references of previous studies, and clarified in “Discussion” that our study presents greater strength than previous studies regarding genetic characterization. 
  • Congratulations to the authors for improving the discussion and considering the comments made in my previous revision of the manuscript. Despite some of the comments not being solved in the first part of the manuscript, now the discussion seems more complete. The improvement of the conclusions has saved the principal flaw of the manuscript and is much more centered on the genetic aspects, which are supported by the data presented. The improvement in the discussion leads now to a general sense of contradiction, as the first part of the manuscript (especially the introduction) makes emphasis on the reintroduction of bottlenose dolphins (with references to the legal status of the dolphins in Mexico which is still not clearly supported by the references cited). In the discussion now is clear that the reintroduction of these individuals is highly unlikely which collides with the arguments provided in the introduction.

In my opinion, this contradiction could be easily solved by avoiding the references to the potential reintroduction of the individuals that were used for the genetic analysis and their legal status in the introduction. The proposed technique can be really useful for the reintroduction or translocation of bottlenose dolphins from a general perspective wherever is necessary for conservation needs (not for administrative or political reasons). For example, it will be very useful for small isolated populations of subspecies like Tursiops truncatus gephyreus (southern Brazil) to have a sensitive genetic tool in order to supplement or translocate individuals from nearby populations. There is no need to particularize this specific group of animals (the subjects of the study) in terms of reintroduction because that makes the argument much weaker and leads to a contradiction. In terms of conservation biology, the methodology also can be much more useful when applied to other odontocete species that will be in need of population management in the next years or decades (Atlantic humpback dolphin (Sousa teuszii), franciscana (Pontoporia blainvillei) or other river dolphins Inia sp. or sotalia sp.)

b) Line 18  Just referring to the use of dolphins under human care for entertainment and assisted therapy does not seem a complete description. See for example:

* Miller, L. J., Zeigler-Hill, V., Mellen, J., Koeppel, J., Greer, T., & Kuczaj, S. (2013). Dolphin Shows and Interaction Programs: Benefits for Conservation Education? Zoo Biology, 32(1), 45–53. https://doi.org/10.1002/zoo.21016

* Harley, H. E., Fellner, W., & Stamper, M. A. (2010). Cognitive Research with Dolphins ( Tursiops truncatus ) at Disney ’ s The Seas : A Program for Enrichment , Science , Education , and Conservation. International Journal of Comparative Psychology, 23, 331–343.

* Cunningham-Smith, P., Colbert, D. E., Wells, R. S., & Speakman, T. (2006). Evaluation of Human Interactions with a Provisioned Wild Bottlenose Dolphin (Tursiops truncatu) near Sarasota Bay, Florida, and Efforts to Curtail the Interactions. Aquatic Mammals, 32(3), 346–356. https://doi.org/10.1578/AM.32.3.2006.346

* O’Brien, J. K., & Robeck, T. R. (2010). The Value of Ex Situ Cetacean Populations in Understanding Reproductive Physiology and Developing Assisted Reproductive Technology for Ex Situ and In Situ Species Management and Conservation Efforts. International Journal of Comparative Psychology, 223(2009), 227–248.

* Fahlman, A., Brodsky, M., Miedler, S., Dennison, S., Ivančić, M., Levine, G., … Borque-Espinosa, A. (2019). Ventilation and gas exchange before and after voluntary static surface breath-holds in clinically healthy bottlenose dolphins, Tursiops truncatus. The Journal of Experimental Biology, 222(5), jeb192211. https://doi.org/10.1242/jeb.192211

* Clegg, I. L. K., & Delfour, F. (2018). Cognitive judgement bias is associated with frequency of anticipatory behavior in bottlenose dolphins. Zoo Biology, 37(2), 67–73. https://doi.org/10.1002/zoo.21400

* Lima, A., Sébilleau, M., Boye, M., Durand, C., Hausberger, M., & Lemasson, A. (2018). Captive Bottlenose Dolphins Do Discriminate Human-Made Sounds Both Underwater and in the Air. Frontiers in Psychology, 9(January), 55. https://doi.org/10.3389/fpsyg.2018.00055

* Giménez, J., Ramírez, F., Almunia, J., G. Forero, M., & de Stephanis, R. (2016). From the pool to the sea: Applicable isotope turnover rates and diet to skin discrimination factors for bottlenose dolphins (Tursiops truncatus). Journal of Experimental Marine Biology and Ecology, 475. https://doi.org/10.1016/j.jembe.2015.11.001

  • We added the four mentioned references, which are now references 57, 58, 59, 60.
  • The comment was not just suggesting the addition of references to the manuscript, in fact, the provided references were supporting the argument that dolphins in captivity are not just used for human entertainment and assisted therapy, but they are used also for education and research on welfare, cognition, physiology, etc. In order to be balanced, a complete description of the use of dolphins in zoo settings must include “education and research”

c) Lines 19-22 Genetic tools are critical to keeping genetic diversity in a captive population but they are not useful for control and identification of the specimens, dolphins are clearly identified by visual cues which is much more cost-effective for specimen identification than genetic markers. 

  • In accordance with this observation, we included a brief explanation in discussion section (page eight, third paragraph, Discussion section). Photo-identification is a technique used in Mexico to identify dolphins in captivity; however, in this work we propose the genetic identification of dolphins as a support in the control of the captive population to avoid impersonation and illegal trafficking of specimens. The photo-identification technique is widely used around the world. However, it has been reported that there is variability in the quality of the photographs, which implies a bias in the identification and also requires specialized software.
  • I understand the advantages of genetics in animal identification to avoid impersonation or illegal trafficking but that need should be supported with scientific evidence of the impersonation or illegal trafficking of the species in Mexico. In other ways, the use of this technique wouldn’t be advantageous compared to simple visual identification based on permanent marks or RFID microchips. Alternatively, the authors can state that this technique can be useful to avoid impersonation and illegal trafficking of specimens in other countries.

d) Line 30 Use reintroduction instead of reinsertion; Line 45 Use reintroduction instead of reinsertion

  • The change was made and homogenized throughout the manuscript.

e) Line 51 The Society for Marine Mammalogy’s taxonomic committee recognizes 37 species.

  • We made the suggested changes in “Introduction section” and include the reference 1. 

f) Lines 59-60 The provided reference does not support the prohibition on the use of marine mammals stated in the text. In fact, it does not even refer to aquatic or marine mammals, cetaceans, or even dolphins.

  • Thank you for this suggestion, it must be a mistake. The references reinforcing the banned of marine mammals in Mexico have been added to the text.
  • Line 61-62 I am afraid the provided reference still does not support the prohibition on the use of marine mammals stated in the text. The current reference is an initiative filed in 2016, six years after that initiative there must be a consolidated text of the Ley General de Vida Silvestre. Consequently, the only reference that can support the text is the Ley General de Vida Silvestre with the modified article 60 Bis.

g) Lines 60-62 The authors suggest that the whole population of captive dolphins in Mexico should be “reinserted” (reintroduced) to the wild, but this is not supported by any reference, please support this with the necessary reference. 

  • We thank the reviewer for this opportune observation. Nonetheless, this is a comment on a personal basis as a consequence of the decision that have been taken regarding the new laws in Mexico. Thus, this comment is not supported by a reference with dolphins. Howewer, in concordance with this pertinent suggestion, we now present a conscientious discussion of the role of these markers in the reintroduction of species and their limitations and low viability of this process in T. Truncatus.
  • Lines 63-65The prohibition of the use of dolphins does not imply the reintroduction of individuals to the sea. In fact, the proposition to change the Ley General de Vida Silvestre in 2016 does not prohibit the use of cetaceans for research and education. Even if the holding of cetaceans were prohibited in the future, the animals could be moved to other countries, or even euthanized. Moreover, the IUCN’s guidelines on animal translocations consider many aspects apart from the genetics (situation of the wild populations, spreading of pathogens, availability of resources, etc.). Hence, currently, the reintroduction of these dolphins is highly unlikely and probably that’s the reason why it is not supported by any scientific reference. Thus citation 7 does not support the statement that these animals “should” be used for reintroduction.

h) Lines 63-84 Despite the authors referring to the need for this study for conservation and reinsertion (AKA reintroduction) it remains unclear how this new set of markers is going to be useful in the conservation of a species considered of Least Concern (Non endangered) by the International Union for the Conservation of Nature. The IUCN’s reintroduction specialist group does not support reintroduction of non-endangered species, as it would not be any conservation benefit and it could be detrimental to the species. This must be clarified and supported with scientific references.

  • We thank the reviewer for this opportune observation. We totally agree with respect to that the main limitation of T. truncatus reintroduction. Following this pertinent observation, we highlight in “Discussion” the importance of the use of this set of genetic markers, we now clearly describe the implications of this set of genetic analysis. Although the dolphin T. truncatus is classified as "Least Concern" there are populations that may be at risk due to small population size, high degree of site fidelity, genetic isolation, and human activities such as chemical and noise pollution, overfishing, accidental entanglement, poaching, and maritime traffic. It is known that vulnerability may be higher in certain regions such as the Caribbean that deserve more attention.

i) Figure 1. ¿Does the distribution of the dolphins by state add any significant information the publication?

  •  This figure showed the distribution of eighteen recreational centers in Mexico. We included a footnote with the meaning of the description of Figure 1.
  • As this information (distribution of the dolphin in facilities by state) is not used in the manuscript to illustrate relevant information on the research I find it irrelevant and distracting. I would not include it on the paper.

j) Lines 114-115 The authors mention that 11 of the 15 strings used for the analysis were described in the previously published analysis, this should be explained in more detail and the previous studies clearly cited, not just included in the supplementary material.

-  In concordance with this pertinent suggestion, we now present all previous published works in Table 1.

- Table 1 does not include the references to the previous studies, the discussion and analysis of the previous work are very poorly addressed. A manuscript that presents a methodology that is supposed to provide the highest discrimination capability can not just explain that in one phrase (lines 247 - 248) at the discussion. A more detailed comparison between the previous methodologies with the one proposed in this manuscript would be much more useful and illustrative to the readers.

k) Table: For clarity table, S2 should be included in the methods section of the manuscript and not in the supplementary materials.

  • In accordance with this observation, we include Table 1 in the text (before Supplementary Table 2).

l) Lines 200 Reference [7] is a study on subjective personality in dolphins, it does not have anything to do with the statement

  • Proper references were added in “Introduction” and the sentence was rewritten.

m) Lines 201-214 The discussion does not add any substantial information on how this methodology is going to help the conservation of the wild populations of dolphins in Mexico, nor how it can be applied to gain relevant information on the genetic structure of the wild populations of dolphins.

  • In concordance with this pertinent suggestion, we now present a detailed description of genetic structure of the Mexican captive T. Truncatus and its implications. 

n) Lines 221 How can it be used in reintroduction programs of a non-endangered species according to the IUCN?

  • Following this meaningful observation, we have included a conscientious discussion of the use of genetic markers in reintroduction, and the limitation of this process in a  non-endangered species according to the IUCN.

o) Lines 226-228 To make such a general assumption a detailed discussion on the management of the captive population in Mexico should be done. This inbreeding situation is not the case in other dolphin captive populations (EEUU or Europe for example), hence it could be the result of a particular population management problem in the country.

  • Indeed, the current state of management of dolphins in Mexico is not compared with other populations, in addition to the new legislation, so the current state of management of dolphin recreational centers in our country is mentioned in the text.

 p) Lines 258-260 The conclusion is not supported by the data presented.

  • Following this meaningful observation, we have included an appropriate conclusion of the study (see Conclusion, ultimate paragraph).

Lines 244-246: This statement is clearly different than the one made in lines 61-62 and is also not supported by the citation. Please include the consolidated text of the Ley General de Vida Silvestre

line 299 The subspecies should be fully named the first time Tursiops truncatus adunctus ; Tursiops truncatus truncatus (and subsequently names can be abbreviated Tursiops t. adunctus ; Tursiops t. truncatus)

Round 3

Reviewer 3 Report

Thank you for your effort in taking into account my suggestions.

Author Response

Dear Editor,

Many thanks for your comments and suggestions. In this setting, we have accepted and realised all the changes suggested by you, which appear in blue font in this new version. Briefly, we removed the italics in the scientific classification of the Delphinidae family and added the author and year of the Tursiops truncatus species, appearing such as Tursiops truncatus, Montagu, 1821. In addition, we add a blank space between numbers and units in the whole text. We took special care in that all species names appear in italics. Regarding your comments about the “subspecie” aduncus, we apologise given that we had two mistakes: 1) Tursiops aduncus is a specie, as you pointed out; thus, we did not consider adding any reference around this point; 2) aduncus is the correct for to write it, which has been homogenised in the text. Finally, we have reviewed our manuscript conscientiously and believe it could be ready for publication; now, the figures present a significant resolution, and all of them have been included in the text.

On behalf of my collaborators and my own appreciate all the reviewer’s suggestions and yours.

Kind regards,

Jonathan Magaña, M.Sc., Ph.D.